# Hypoxic Processes Induce Complement Activation via Classical Pathway in Porcine Neuroretinas

**DOI:** 10.3390/cells10123575

**Published:** 2021-12-18

**Authors:** Ana M. Mueller-Buehl, Torsten Buehner, Christiane Pfarrer, Leonie Deppe, Laura Peters, Burkhard H. Dick, Stephanie C. Joachim

**Affiliations:** 1Experimental Eye Research Institute, University Eye Hospital, Ruhr-University Bochum, 44892 Bochum, Germany; Ana.Mueller-Buehl@rub.de (A.M.M.-B.); tb@tierarzt-buehner.de (T.B.); Leonie.Deppe@rub.de (L.D.); Laura.Peters@rub.de (L.P.); Burkhard.Dick@kk-bochum.de (B.H.D.); 2Institute for Anatomy, University of Veterinary Medicine Hannover, 30559 Hannover, Germany; Christiane.Pfarrer@tiho-hannover.de

**Keywords:** hypoxia, oxidative stress, complement system, microglia, inflammation

## Abstract

Considering the fact that many retinal diseases are yet to be cured, the pathomechanisms of these multifactorial diseases need to be investigated in more detail. Among others, oxidative stress and hypoxia are pathomechanisms that take place in retinal diseases, such as glaucoma, age-related macular degeneration, or diabetic retinopathy. In consideration of these diseases, it is also evidenced that the immune system, including the complement system and its activation, plays an important role. Suitable models to investigate neuroretinal diseases are organ cultures of porcine retina. Based on an established model, the role of the complement system was studied after the induction of oxidative stress or hypoxia. Both stressors led to a loss of retinal ganglion cells (RGCs) accompanied by apoptosis. Hypoxia activated the complement system as noted by higher C3^+^ and MAC^+^ cell numbers. In this model, activation of the complement cascade occurred via the classical pathway and the number of C1q^+^ microglia was increased. In oxidative stressed retinas, the complement system had no consideration, but strong inflammation took place, with elevated *TNF*, *IL6*, and *IL8* mRNA expression levels. Together, this study shows that hypoxia and oxidative stress induce different mechanisms in the porcine retina inducing either the immune response or an inflammation. Our findings support the thesis that the immune system is involved in the development of retinal diseases. Furthermore, this study is evidence that both approaches seem suitable models to investigate undergoing pathomechanisms of several neuroretinal diseases.

## 1. Introduction

For all research fields it is inevitable continue looking for alternative ex vivo or in vitro models, where animal models can be replaced or at least reduced. Of course, the background of the investigated diseases should remain the focus and models must be suitable for the analyses of undergoing pathomechanisms.

Retinal diseases, such as glaucoma, age-related macular degeneration (AMD), or diabetic retinopathy, are multifactorial, and hypoxia as well as oxidative stress seem to be involved. It is known that both stressors play an important role in the progression of several retinal diseases [1,2,3,4]. For almost all glaucoma forms, an increased intraocular pressure (IOP) is the main risk factor and current glaucoma treatment targets the IOP. It is known that oxidative stress and hypoxia are pivotal in the progression of several retinal diseases [5,6,7]. Through a higher amount of unsaturated fatty acids and high oxygen consumption, neuronal cells are more vulnerable to oxidative damage than other cells, which leads to an early degeneration of neuronal cell types within the retinal tissue [8,9,10].

Several studies have examined the role of other possible pathogenic factors for glaucoma, such as excitotoxicity, circulatory disorders, oxidative stress, hypoxia, and a dysregulated immune system [11,12,13,14]. In secondary glaucoma forms, neovascular glaucoma (NGV), inflammation processes as well as elevated levels of vascular endothelial growth factor (VEGF) in the eye are main characteristics. Anti-VEGF agents act against the inflammation via the PLA2/COX-2/VEGF-A pathway [15]. Hence, these agents can be used to treat NGV, but they are not yet an approved therapy [2].

The complement system, as part of the innate immune system, is gaining importance in enabling understanding of the emergence of retinal diseases, such as glaucoma or AMD [14,16,17,18]. It is evidenced that the complement system is dysregulated in AMD and that the drusen, which are characteristic for AMD, contain complement components [19]. The complement system is a complex network of more than 50 soluble and surface proteins, which are activated through proteolytic cascades [16]. Activation of the complement cascade can occur via three different ways: the classical, the lectin, and the alternative pathway [20]. The classical pathway is induced by the binding of complement component 1 (C1), especially its subunit C1q, to an antibody marked pathogen [21]. In the lectin pathway, mannose-binding lectin (MBL) binds to certain sugars on the pathogens surface, whereas in the alternative pathway, the pathogen surface itself activates the complement cascade [22]. All three routes lead to the building of C3-convertase which then ends in the cell lysis induced by terminal membrane attack complex (MAC). Inappropriate activation of the complement system is known to play a crucial role in many neurodegenerative diseases, such as multiple sclerosis [23] and Alzheimer’s disease [24,25]. Upregulated levels of complement proteins were also found in aqueous humour, sera, and retinas of patients with primary open-angle glaucoma [13,17,18,26,27]. Even if previous studies examined the role of the complement system in retinal diseases, the exact role is still unclear. To gain a better insight, we used porcine retinas to investigate the role of the complement system in hypoxia- and oxidative stress-induced retinal damage.

For ophthalmic research, several rodent animal models have been established. However, rodents might not be the best model to mimic human diseases. Besides the diverging size of the eyes, rodents are nocturnal mammals with excellent (scotopic) night-vision [28]. Therefore, the distribution of photoreceptors in the retina is quite different to humans. Rodents do not have a macula, in contrast to humans, or a visual stripe, like pigs. They have a flat eyeball compared to the almost spherical shape of the human eye. An established alternative model for retinal research is the organ culture model of porcine retina [16,29,30]. Porcine eyes can be obtained from local slaughterhouses as a waste product of the food industry. Therefore, our model is a compensation of conventional animal testing, which helps to reduce the number of animals for testing. The purchase of pig eyes does not require the killing of animals solely for research. Moreover, the anatomy of pig eyes and its retinas is much more like the one of humans. Pig’s vision is dichromatic, due to two opsins. Furthermore, the density of rods and cones in porcine retinas is more human-like than in rodent retinas [31]. The major benefit of an ex vivo model is the possibility to reduce the number of laboratory animals. Even with cell cultures being helpful in examining many pathways, the complexity of an organ culture can make us understand the processes within a whole organ, since the connection between different cell types is still given.

Several pathomechanisms can be induced ex vivo, to mimic neuroretinal damage [6,7]. Applying hydrogen peroxide (H_2_O_2_) to a porcine ex vivo model leads to building of O_3_ within the tissue [7]. Through this process, reactive oxygen species are produced, which in sufficient amount leads to oxidative stress and thereby to a degeneration within the retina [7]. Furthermore, cobalt chloride (CoCl_2_) can be used to obtain a damage mimicking hypoxia [6]. Hypoxia is a poor perfusion of the tissue, which leads to an upregulation of the transcription factor hypoxia inducible factor-1 (HIF-1) and the stabilization of its subunit HIF-1α. Then, HIF-1α is translocated into the cell nucleus, where hypoxic genes are expressed [32,33,34]. In addition, CoCl_2_ disturbs the mitochondrial respiration by rupturing the cell membrane, which reinforces the degenerative effect on the tissue [29]. Furthermore, CoCl_2_ can modulate not only HIF-1α but also VEGF and placental growth factor (PlGF) [35].

In the present study, we examined the role of the complement system and microglia in hypoxic and oxidative stressed cultured porcine retinas. Our results indicate that hypoxia triggers the activation of the complement cascade in this ex vivo retina model via the classical pathway. In addition, C1q^+^ microglia might contribute to retinal damage.

## 2. Materials and Methods

### 2.1. Preparation and Cultivation of Porcine Retinal Explants

Porcine eyes were obtained from the local slaughterhouse. The preparation of retinal explants was performed as described previously [6,7,29,36,37]. Briefly, anterior parts of the eye, including cornea, pupil, lens, and the vitreous body were removed. The remaining eyecup was cut into four parts to get a shamrock like structure. One retinal explant, located in the center of each quarter, was punched out with a biopsy punch (Ø = 6 mm, Pfm medical AG, Cologne, Germany) and placed on an insert (Merck Millipore, Darmstadt, Germany) in a 6-well plate with the photoreceptor layer facing the filter. Each well was filled with 1 mL Neurobasal-A-medium (Thermo Fisher, Waltham, MA, USA) supplemented with 0.8 mM L-glutamine (Thermo Fisher), 2% B27 (Thermo Fisher), 1% N2 (Thermo Fisher), and 2% penicillin/streptomycin (Sigma–Aldrich, St. Louis, MO, USA).

Retinal explants were cultivated for two and four days (n = 14/group/point in time). Degeneration was induced by adding 500 µM H_2_O_2_ or 300 µM CoCl_2_ at day one for three hours, as previously established [6]. Explants cultivated without any stressors served as controls. In all groups, medium was replaced on day zero, one, and three (Figure 1). After cultivation, retinas later used for (immuno-)histology (n = 8/group/point in time) were fixed in 4% PFA for 15 min followed by dehydration using 15% and 30% sucrose diluted in PBS for 15 or 30 min respectively. Retinas used for RT-qPCR (n = 4–5/group/point in time) were frozen at −80 °C without any further preparation. Retinas for spectral-domain optical coherence tomography (SD-OCT) were also used without modification.

### 2.2. Spectral-Domain Optical Coherence Tomography (SD-OCT)

SD-OCT analysis of porcine retinas was performed on day two and four using an SD-OCT (Spectralis, Heidelberg Engineering, Heidelberg, Germany). As described before, a special adaptor was used with a customized mounting device, which fits the filter inserts (Ø 12 mm) [38]. For analysis of retinal thickness, three 30°-line scans (ART:100)-images of each retina were taken. For each image, five measurements were performed for the total retinal thickness as well as for the retinal ganglion cell complex, consisting of the ganglion cell layer (GCL), the inner plexiform layer (IPL), and the inner nuclear layer (INL). In total, 15 measurements of each retina, with n = 5–6/group/point in time, were performed as described.

### 2.3. (Immuno-) Histological Staining of Retinal Sections

For (immuno-) histological staining, two slides per retina were used. Each slide contains three retinal cross-sections of 10 µm thickness. In total, for each staining, six cross-sections per retina were analyzed. We used n = 8/group/point in time for (immuno-) histological staining [39]. Retinas were stained with hematoxylin and eosin (H and E) using established protocols to measure the thickness of retinal layers via ZEN 2012-imaging-software (blue edition, ZEISS, Oberkochen, Germany) [8,40]. Therefore, we took two photos from each retina section and measured the thickness of the whole retina in three areas (excluding photoreceptor layers). In total, 12 images per retina were included for evaluation of retinal thickness. The mean measurement for each retina was used for the statistical analysis.

To analyze the influence of chemically induced neurodegeneration on RGCs and apoptosis (NeuN, TuJ1, cl. caspase 2), the complement system (C3, MAC, C1q, MASP2, factor B) as well as microglia (Iba1), different antibodies were applied for immunofluorescence (Table 1). For staining, slides were dried and rehydrated in PBS, before 1 h blocking in a blocking buffer containing 10–20% normal donkey serum, 10% normal goat serum, 1–10% bovine serum albumin, and 0.1–0.2% TritonX solved in PBS. Slides were incubated with primary antibodies diluted in blocking buffer at room temperature overnight. The next day, respective secondary antibodies were diluted in the same blocking buffer and incubated for one hour at room temperature (Table 1). Cell nuclei were visualized with 4′,6′-Diamidin-2-Phenylindol (DAPI, Dianova, Hamburg, Germany).

For the evaluation of the cell numbers, four images of each retinal-cross section were taken with a fluorescence microscope (Axio Imager M1, Zeiss). In total, 24 images were taken and analyzed for each retina. After microscopy, images were cropped with a defined window size (800 × 600 pixel) and then masked for cell countings. Stainings for NeuN, cl. caspase 2, C3, C5, C1q, Iba1, factor B, and MAC were evaluated by cell counting with ImageJ 1.47v-software. Evaluations of cell counts for C3, MAC, and factor B were separated into immunopositive cells located in the GCL as well as in the INL + IPL. Those counts resulted in the total amount of immunopositive cells in the retina.

Cells were only counted as positive when the respective signal was co-localized with DAPI. For Iba1-C1q co-staining, Iba1 signals co-localized with DAPI were counted as microglia and C1q^+^ and Iba1^+^ signals co-localized with DAPI were counted as C1q^+^ microglia. For the conversion into cells per mm, we first measured the length of the cropped image (in mm) and the length of the scale bar (in mm; = x). With the following equations, a factor (y) was calculated, which was used to convert the total cell count into mm.
(1)$$x=\frac{\mathrm{lenght}\;\mathrm{of}\;\mathrm{the}\;\mathrm{cropped}\;\mathrm{image}\;\left(\mathrm{mm}\right)}{\mathrm{lenght}\;\mathrm{of}\;\mathrm{the}\;\mathrm{scale}\;\mathrm{bar}\;\left(\mathrm{mm}\right)}$$
(2)$$y=\frac{X\times \mathrm{cale}\;\mathrm{bar}\;\left(e.g.,\;20\;µm\right)}{1000}$$
(3)$$\mathrm{cells}\;[\mathrm{mm}]=\frac{\mathrm{counted}\;\mathrm{cell}\;\mathrm{number}}{y}$$

The MASP2 and TuJ1 positive marked area was measured using an ImageJ macro (NIH), as described before [41]. Briefly, signals were transferred into 32-bit grey scale and background was subtracted (MASP2 and TuJ1: 50 pixel). Lower and upper threshold of each image were determined. The mean value for lower and maximum from upper threshold of all pictures was used for further analyses (MASP2: lower threshold: 8.42; upper threshold: 85.53; TuJ1: lower threshold: 13.87; upper threshold: 104.49). Results of MASP2 and TuJ1 are presented as positive signal (in %) per cropped image.

### 2.4. RNA Isolation and cDNA Synthesis

Retinas were lysed for RNA isolation using lysis buffer containing 2-mercaptoethanol (Sigma-Aldrich). RNA extraction was done according to the manufacture’s instruction using Gene Elute Mammalian Total RNA Miniprep Kit. RNA concentration was measured by NanoDropTM One (Thermo Fisher) spectrophotometer. 1 µg RNA was used for cDNA synthesis with First Strand cDNA Synthesis Kit (Thermo Fisher).

### 2.5. Quantitative Real-time PCR (RT-qPCR)

RT-qPCR analyses were carried out with SYBR Green I protocol on PikoReal 96 (Thermo Fisher) [42]. A plate layout with duplicates for each cDNA and primer was created in advance. Suitable sequences of nucleotides were searched in PubMed and primers were blasted with PubMed tool. On 96-well plates a final volume of 20 µL/well (5 µL cDNA mix and 15 µL Primermix, Thermo Fisher) was applied. Every analyzed gene was normalized to the housekeeping genes *β-ACTIN* (*ACTB*) and *Histocompatibility 3* (*H3*; Table 2). Ct-values of each sample were analyzed with PikoReal 2.2 software (Thermo Fisher). By using REST software (Qiagen, Hilden, Germany), significances between stressed groups and the control group were evaluated.

### 2.6. Statistical Analyses

All groups were compared by one-way ANOVA, followed by Tukey’s honest post-hoc test for equal groups (Statistica V13; Palo Alto, CA, USA). Results of immunohistological analyses are presented as mean ± SEM. RT-qPCR were analyzed with REST Software (Qiagen) and analyses are shown as mean ± SD ± SEM. For RT-qPCR analyses, the H_2_O_2_ and the CoCl_2_ group are always displayed in relation to the control group. The control group was set at 1 and is represented by a dotted line in all graphs. Statistically significant was a result with a *p*-value under 0.05 with *, ^#^: *p* < 0.05, **, ^##^: *p* < 0.01, and ***, ^###^: *p* < 0.001. Significant differences in comparison to the control group are marked with a *, significant differences within the H_2_O_2_ and CoCl_2_ groups are labeled with a ^#^.

## 3. Results

### 3.1. Reduction of Retinal Thickness in Cobalt Chloride Treated Retinas

SD-OCT allows the analysis of the retinal thickness of cultured retinas during the cultivation (Figure 2A). At two days, retinal thickness was comparable in all three groups (control: 221.67 ± 5.80 µm; H_2_O_2_: 208.28 ± 3.16 µm, *p* = 0.273; CoCl_2_: 200.18 ± 7.63 µm; *p* = 0.055; Figure 2B). Moreover, the measurement of the ganglion cell complex, containing the GCL, IPL, and INL did not reveal any differences within the groups (control: 87.38 ± 6.21 µm; H_2_O_2_: 82.05 ± 5.34 µm, *p* = 0.788; CoCl_2_: 75.09 ± 5.48 µm; *p* = 0.307; Figure 2C). At four days, retinal thickness of H_2_O_2_-stressed retinas was not altered in comparison to control ones (control:218.85 ± 5.61 µm; H_2_O_2_: 207.52 ± 3.02 µm; *p* = 0.289). However, in CoCl_2_ retinas, a significant reduction of retinal thickness was noticed (196.73 ± 6.02; *p* = 0.024; Figure 2D). Oxidative stress had no effect on the thickness of the ganglion cell complex (control: 87.00 ± 2.81 µm; H_2_O_2_: 77.40 ± 4.07 µm, *p* = 0.207). Interestingly, the ganglion cell complex of CoCl_2_ stressed retinas was significantly thinner than of control samples (69.31 ± 4.20 µm; *p* = 0.015; Figure 2E).

Additionally, H & E staining was performed to measure the total retinal thickness of retinal cross-sections (Figure 3A). After two days, no differences were seen in the retinal thickness neither in oxidative stressed nor in hypoxic retinas in comparison to the control group (control: 97.00 ± 5.94 µm; H_2_O_2_: 85.76 ± 3.27 µm, *p* = 0.154; CoCl_2_: 84.67 ± 2.17 µm; *p* = 0.109; Figure 3B). While the retinal thickness was not affected by oxidative stress at day four (control: 90.06 ± 2.25 μm; H_2_O_2_: 83.64 ± 1.68 μm, *p* = 0.151), hypoxia via CoCl_2_ led to a significantly reduced retinal thickness (78.63 ± 2.90 μm; *p* = 0.006; Figure 3C).

To investigate the damage of RGCs in cultured stressed porcine retinas, we used TuJ1 as an additional marker for RGCs (Figure 3D). After two days, the immunopositive area of TuJ1 was not altered in any of the groups (control: 18.16 ± 1.13% TuJ1^+^ area; H_2_O_2_: 13.73 ± 1.73% TuJ1^+^ area, *p* = 0.111; CoCl_2_: 14.10 ± 1.42% TuJ1^+^ area, *p* = 0.152; Figure 3E). However, after four days, a significantly reduced TuJ1^+^ area was noted in CoCl_2_-stressed retinas in comparison to the control group (control: 18.59 ± 3.62% TuJ1^+^ area; H_2_O_2_: 17.14 ± 1.84% TuJ1^+^ area, *p* = 0.903; CoCl_2_: 8.88 ± 0.79% TuJ1^+^ area, *p* = 0.029; Figure 3E).

### 3.2. Early Loss of RGCs Due to Apoptotic Mechanisms

To investigate the apoptotic stage of RGCs in H_2_O_2_- and CoCl_2_-stressed retinas, neuronal retinal cells were stained with NeuN and apoptotic cells with cleaved caspase 2 (Figure 4A). At two days, a significant loss of NeuN^+^ cells was noted after oxidative stress (control: 52.94 ± 5.81 NeuN^+^ cells/mm; H_2_O_2_: 36.61 ± 1.82 NeuN^+^ cells/mm, *p =* 0.039). The number of NeuN^+^ cells was not affected through CoCl_2_ (43.49 ± 3.74 NeuN^+^ cells/mm; *p =* 0.275; Figure 4B). At two days, cell counts of apoptotic NeuN^+^ cells revealed no statistical differences between the three groups (control: 44.85 ± 8.47 cl. caspase 2^+^ and NeuN^+^ cells [%]; H_2_O_2_: 43.76 ± 2.35 cl. caspase 2^+^ and NeuN^+^ cells [%], *p =* 0.989; CoCl_2_: 45.79 ± 3.38 cl. caspase 2^+^ and NeuN^+^ cells [%], *p =* 0.992; Figure 4C).

After four days of cultivation, oxidative stress (38.32 ± 1.25 NeuN^+^ cells/mm; *p* < 0.001) as well as hypoxic processes (42.12 ± 2.45 NeuN^+^ cells/mm; *p =* 0.011) led to significantly reduced NeuN^+^ cell counts compared to control samples (51.08 ± 2.11 NeuN^+^ cells/mm; Figure 4D). Interestingly, the number of apoptotic NeuN^+^ cells was increased in H_2_O_2_-treated retinas after four days, even though this effect was not significant (control: 43.86 ± 2.36 cl. caspase 2^+^ and NeuN^+^ cells [%]; H_2_O_2_: 52.98 ± 3.63 cl. caspase 2^+^ and NeuN^+^ cells [%], *p =* 0.106). However, loss of NeuN^+^ cells due to CoCl_2_ was accompanied by a significantly increased apoptosis rate (55.31 ± 2.99 cl. caspase 2^+^ and NeuN^+^ cells [%]; *p =* 0.035; Figure 4E).

Additionally, RT-qPCR analyses were performed to investigate the mRNA expression of *TUBB3*, *BAX*, and *BCL2*. Relative mRNA levels of *TUBB3* were not altered after two days (H_2_O_2_: 1.05-fold expression, *p =* 0.835; CoCl_2_: 1.27-fold expression; *p =* 0.073), but significantly downregulated after four days (H_2_O_2_: 0.76-fold expression, *p =* 0.020; CoCl_2_: 0.42-fold expression; *p =* 0.022, Figure 4F). At two days, *BAX* mRNA levels were neither affected by H_2_O_2_ (1.18-fold expression; *p =* 0.792) nor by CoCl_2_ (1.06-fold expression; *p =* 0.936). After four days, mRNA level of H_2_O_2_-stressed retinas was 7.94-fold higher than in control retinas, but this effect was still not significant (*p =* 0.087). Also, mRNA levels of *BAX* were not altered through CoCl_2_ (0.60-fold expression; *p =* 0.615; Figure 4G). Another gene that was evaluated was the pro-apoptotic gene *BCL2*. Oxidative stress due to H_2_O_2_ led to a significantly reduced mRNA level of *BCL2* at two days (0.029-fold expression, *p =* 0.002). CoCl_2_, however, did not affect the mRNA expression level of *BCL2* (1.5-fold expression, *p =* 0.266). At four days, the expression of *BCL2* was still lower in H_2_O_2_-stressed retinas, but this effect was not significant (0.64-fold expression, *p* = 0.241). Interestingly, *BCL2* mRNA level was significantly upregulated in hypoxic retinas (1.78-fold expression, *p* = 0.030; Figure 4H).

### 3.3. Increase of C3^+^ Cells by Hypoxia in Total Retina

C3, an important factor in the complement cascade, was examined histologically at day two and four (Figure 5A). Amongst other functions, C3 serves the opsonization of the target cell. After two days, similar numbers of C3^+^ cells were found in the GCL of all groups (control: 4.49 ± 0.66 C3^+^ cells/mm; H_2_O_2_: 5.36 ± 0.56 C3^+^ cells/mm, *p* = 0.776; CoCl_2_: 6.77 ± 1.31 C3^+^ cells/mm, *p* = 0.199; Figure 5B). The number of C3^+^ cells in the inner retinal layers (IPL + INL) was also comparable in all groups (control: 5.62 ± 0.91 C3^+^ cells/mm; H_2_O_2_: 6.53 ± 0.75 C3^+^ cells/mm, *p* = 0.736; CoCl_2_: 8.67 ± 0.92 C3^+^ cells/mm, *p* = 0.052; Figure 5C). On the other hand, the evaluation of C3^+^ cells the total retina showed no altered number of C3^+^ cells in the H_2_O_2_ group and a significantly increased number in CoCl_2_ group (control: 10.11 ± 0.73 C3^+^ cells/mm; H_2_O_2_: 11.9 ± 1.03 C3^+^ cells/mm, *p* = 0.595; CoCl_2_: 15.44 ± 1.84 C3^+^ cells/mm, *p* = 0.021; Figure 5D).

At four days, no changes were observed regarding the number of C3^+^ cells located in the GCL (control: 5.04 ± 1.10 C3^+^ cells/mm; H_2_O_2_: 3.11 ± 0.27 C3^+^ cells/mm, *p* = 0.121; CoCl_2_: 3.23 ± 0.19 C3^+^ cells/mm, *p* = 0.153; Figure 5E). Again, at four days C3^+^ cell counts in the inner retinal layers were not altered within the groups (control: 2.90 ± 0.58 C3^+^ cells/mm; H_2_O_2_: 2.30 ± 0.34 C3^+^ cells/mm, *p* = 0.745; CoCl_2_: 3.59 ± 0.75 C3^+^ cells/mm, *p* = 0.686; Figure 5F). In the total retina, there were significantly fewer C3^+^ cells in the H_2_O_2_ group compared to controls (control: 7.99 ± 0.94 C3^+^ cells/mm; H_2_O_2_: 5.40 ± 0.39 C3^+^ cells/mm, *p* = 0.046; Figure 5G), whereas no difference was noted in CoCl_2_ retinas (6.82 ± 0.69 C3^+^ cells/mm, *p* = 0.488; Figure 5G).

### 3.4. Cobalt Chloride Increased the Number of MAC^+^ Cells in Inner Retinal Layers

The final path of the complement cascade was stained by using an antibody against MAC (Figure 6A). MAC leads to the lysis of prior opsonized target cell. At two days, a comparable number of MAC^+^ cells located in the GCL was found in all groups (control: 2.46 ± 0.57 MAC^+^ cells/mm; H_2_O_2_: 2.94 ± 0.53 MAC^+^ cells/mm, *p* = 0.796; CoCl_2_: 2.54 ± 0.49 MAC^+^ cells/mm, *p* = 0.994; Figure 6B). The inner layers of the retina showed no difference in the number of MAC^+^ cells at this point in time (control: 4.72 ± 1.30 MAC^+^ cells/mm; H_2_O_2_: 5.53 ± 0.53 MAC^+^ cells/mm, *p* = 0.841; CoCl_2_: 6.69 ± 1.05 MAC^+^ cells/mm, *p* = 0.369; Figure 6C). Matching the results of the separately counted layers, no statistically significant difference was found in the number of MAC^+^ cells in the total retina at two days (control: 7.18 ± 1.40 MAC^+^ cells/mm; H_2_O_2_: 8.47 ± 0.67 MAC^+^ cells/mm, *p* = 0.692; CoCl_2_: 9.24 ± 1.12 MAC^+^ cells/mm, *p* = 0.402; Figure 6D).

Additionally, after four days, counting of MAC^+^ cells in the GCL showed no differences between groups (control: 2.90 ± 0.58 MAC^+^ cells/mm; H_2_O_2_: 1.81 ± 0.33 MAC^+^ cells/mm, *p* = 0.181; CoCl_2_: 1.77 ± 0.27 MAC^+^ cells/mm, *p* = 0.161; Figure 6E). Interestingly, an increased number of MAC^+^ cells was found in the CoCl_2_ group, but not in the H_2_O_2_ group, in the inner layers of the retina (control: 1.86 ± 0.58 MAC^+^ cells/mm; H_2_O_2_: 2.26 ± 0.43 MAC^+^ cells/mm, *p* = 0.837; CoCl_2_: 3.71 ± 0.47 MAC^+^ cells/mm, *p* = 0.039; Figure 6F). This increase was not seen when counting MAC^+^ cells in the total retina, were the number of MAC^+^ cells was alike in all groups (control: 4.80 ± 0.60 MAC^+^ cells/mm; H_2_O_2_: 4.07 ± 0.30 MAC^+^ cells/mm, *p* = 0.557; CoCl_2_: 5.48 ± 0.52 MAC^+^ cells/mm, *p* = 0.592; Figure 6G).

In accordance with the histological staining, RT-qPCR analyses for *C5* mRNA were performed. The *C5* mRNA expression was not altered by H_2_O_2_ or CoCl_2_ at two days (H_2_O_2_: 0.79-fold expression, *p* = 0.854; CoCl_2_: 0.77-fold expression, *p* = 0.489). This was also the case at four days (H_2_O_2_: 0.48-fold expression, *p* = 0.442; CoCl_2_: 0.45-fold expression, *p* = 0.331; Figure 6H).

### 3.5. No Activation of the Alternative and Lectin Pathway through Hydrogen Peroxide or Cobalt Chloride

The complement cascade can be activated through three different pathways [43]. To identify which pathway plays a role in the organ culture model of oxidative stressed or hypoxic porcine retinas, markers for each pathway were examined. Factor B is the main activator of the alternative pathway [44]. To examine this pathway, histological staining with an antibody for factor B was performed on two and four days of cultivated retinas (Figure 7A). After two days, no differences in the number of factor B^+^ cells in the GCL were found (control: 5.65 ± 1.63 factor B^+^ cells/mm; H_2_O_2_: 6.05 ± 3.00 factor B^+^ cells/mm, *p* = 0.991; CoCl_2_: 5.24 ± 1.60 factor B^+^ cells/mm, *p* = 0.991; Figure 7B). Also, the evaluation of factor B^+^ cells in the inner layers of the retina showed no differences between any of the groups at this point in time (control: 0.60 ± 0.24 factor B^+^ cells/mm; H_2_O_2_: 0.60 ± 0.31 factor B^+^ cells/mm, *p* = 1.000; CoCl_2_: 1.21 ± 0.64 factor B^+^ cells/mm, *p* = 0.591; Figure 7C). Similar results were found in the total retina, where a comparable number of factor B^+^ cells was seen in all groups after two days (control: 6.25 ± 1.63 factor B^+^ cells/mm; H_2_O_2_: 6.65 ± 3.06 factor B^+^ cells/mm, *p* = 0.992; CoCl_2_: 6.45 ± 2.11 factor B^+^ cells/mm, *p* = 0.998; Figure 7D).

At four days, the H_2_O_2_ and the control group revealed similar factor B^+^ cell counts in the GCL (control: 12.87 ± 2.21 factor B^+^ cells/mm; H_2_O_2_: 13.23 ± 2.52 factor B^+^ cells/mm; *p* = 0.991). In contrast, CoCl_2_ stressed retinas displayed significantly fewer factor B^+^ cells (4.63 ± 1.31 factor B^+^ cells/mm) in comparison to controls (*p* = 0.028) as well as to H_2_O_2_ treated retinas (*p* = 0.021; Figure 7E). In contrast, the number of factor B^+^ cells was not altered in the inner layers of the retina at this point in time (control: 1.21 ± 0.42 factor B^+^ cells/mm; H_2_O_2_: 1.61 ± 0.58 factor B^+^ cells/mm, *p* = 0.839; CoCl_2_: 1.49 ± 0.50 factor B^+^ cells/mm, *p* = 0.917; Figure 7F). Results regarding the cell number of factor B^+^ cells in the total retina matched the ones described for the GCL. Again, hypoxic retinas had significantly fewer factor B^+^ cells than control as well as H_2_O_2_ treated ones (control: 14.08 ± 2.25 factor B^+^ cells/mm, *p* = 0.032; CoCl_2_: 6.13 ± 1.31 factor B^+^ cells/mm; H_2_O_2_: 14.84 ± 2.40 factor B^+^ cells/mm, *p* = 0.018; Figure 7G).

In addition, RT-qPCR analyses for *complement factor B* (*CFB*) were performed. All groups showed a comparable *CFB* mRNA expression of after two days (H_2_O_2_: 1.11-fold expression, *p* = 0.654; CoCl_2_: 1.40-fold expression, *p* = 0.204). This was still the case at four days of cultivation (H_2_O_2_: 1.26-fold expression, *p* = 0.748; CoCl_2_: 0.60-fold expression, *p* = 0.542; Figure 7H). Another gene that was investigated was *complement factor H* (*CFH*), a complement system regulator. A significantly upregulated *CFH* expression was found in H_2_O_2_ treated samples at day two (2.37-fold expression, *p* = 0.043), while *CFH* mRNA levels were not altered in hypoxic retinas (1.30-fold expression, *p* = 0.423). Furthermore, after four days, the mRNA expression of *CFH* was significantly upregulated the H_2_O_2_ group (3.50-fold expression, *p* = 0.010). Interestingly, the mRNA expression of *CFH* was 3-fold higher in hypoxic retinas, but this effect was still not significant (2.996-fold expression, *p* = 0.086; Figure 7I).

For examination of the lectin pathway, a MASP2 antibody was used for histological staining (Figure 8A). MASP2 is involved in the activation via the lectin pathway. After two days, the MASP2^+^ area was comparable in all groups (control: 5.14 ± 0.68 MASP2^+^ area [%]/image; H_2_O_2_: 5.55 ± 0.81 MASP2^+^ area [%]/image, *p* = 0.888; CoCl_2_: 4.14 ± 0.30 MASP2^+^ area [%]/image, *p* = 0.517; Figure 8B). Moreover, after four days, no differences regarding the MASP2 area were detected between groups (control: 5.61 ± 0.76 MASP2^+^ area [%]/image; H_2_O_2_: 5.54 ± 0.92 MASP2^+^ area [%]/image, *p* = 0.997; CoCl_2_: 4.00 ± 0.46 MASP2^+^ area [%]/image, *p* = 0.293; Figure 8C).

Complementary to histological staining, RT-qPCR analyses were performed at two and four days of cultivation. We observed no differences regarding *MASP2* mRNA expression of after two days (H_2_O_2_: 1.07-fold expression, *p* = 0.832; CoCl_2_: 1.07-fold expression, *p* = 0.720). At four days *MASP2* expression levels were still similar in all groups (H_2_O_2_: 0.58-fold expression, *p* = 0.284; CoCl_2_: 1.44-fold expression, *p* = 0.480; Figure 8D).

### 3.6. Fewer Microglia but More C1q^+^ Microglia in Hypoxic Retinas

Previous studies show that microglia can produce complement components, such as C1q, and might play a role in the pathogenesis of glaucoma [16,45]. For examination of the connection between microglia and C1q, a double staining with antibodies for Iba1, which labels microglia, and C1q, as a member of the classical pathway of the complement system, was performed (Figure 9A). After two days, there was a significant increase in the number of Iba1^+^ cells in the H_2_O_2_ group compared to the control (control: 22.38 ± 4.63 Iba1^+^ cells/mm; H_2_O_2_: 28.76 ± 3.92 Iba1^+^ cells/mm; *p* = 0.016) and the CoCl_2_ group (CoCl_2_: 20.85 ± 3.92 Iba1^+^ cells/mm, *p* = 0.003; Figure 9B). At four days, effects of oxidative stress due to H_2_O_2_ was not that strong anymore (control: 44.52 ± 1.74 Iba1^+^ cells/mm; H_2_O_2_: 51.90 ± 4.64 Iba1^+^ cells/mm, *p* = 0.330), whereas CoCl_2_ induced a significant loss of microglia (28.76 ± 3.71 Iba1^+^ cells/mm; Figure 9C). This effect was significant in comparison to the control (*p* = 0.014) as well as to the H_2_O_2_ group (*p* = 0.0006).

Accordingly, RT-qPCR analyses of *ITGAM* mRNA expression, which encodes for Cd11b, a microglia specific protein, were performed. At two days, the mRNA expression level of *ITGAM* was not affected by H_2_O_2_ (0.85-fold expression, *p* = 0.630), but significantly reduced by CoCl_2_ (0.40-fold expression, *p* = 0.003). At four days, no effects on *ITGAM* mRNA levels were noted neither through H_2_O_2_ (0.54-fold expression, *p* = 0.249) nor CoCl_2_ (1.03-fold expression, *p* = 0.915; Figure 9D).

Regarding the number of C1q^+^ cells, counts were comparable in all groups at two days (control: 6.01 ± 0.68 C1q^+^ cells/mm; H_2_O_2_: 5.81 ± 0.83 C1q^+^ cells/mm, *p* = 0.983; CoCl_2_: 7.38 ± 0.92 C1q^+^ cells/mm, *p* = 0.473; Figure 9E). Also, after four days, no difference was noted in C1q^+^ cell counts (control: 9.44 ± 0.80 C1q^+^ cells/mm; H_2_O_2_: 9.68 ± 1.28 C1q^+^ cells/mm, *p* = 0.987; CoCl_2_: 10.81 ± 1.17 C1q^+^ cells/mm, *p* = 0.659; Figure 9F).

Additionally, C1q^+^ and Iba1^+^ cells were counted. Thereby, at two days, a significantly increased number of C1q^+^ and Iba1^+^ cells was noted in the CoCl_2_-stressed group compared to the control group (control: 11.15 ± 1.74 C1q^+^ and Iba1^+^ cells [%]; CoCl_2_: 20.07 ± 2.63 C1q^+^ and Iba1^+^ cells [%], *p* = 0.009) as well as to the H_2_O_2_ group (H_2_O_2_: 10.56 ± 0.95 C1q^+^ and Iba1^+^ cells [%]; *p* = 0.005; Figure 9G). Interestingly, also at four days, a significantly higher number of C1q^+^ microglia was found in the CoCl_2_ group, compared to the control group, even though the total number of microglia was lower in the CoCl_2_-stressed group (control: 12.90 ± 1.90 C1q^+^ and Iba1^+^ cells [%]; CoCl_2_: 21.83 ± 1.57 C1q^+^ and Iba1^+^ cells [%], *p* = 0.003; H_2_O_2_: 16.01 ± 1.57 C1q^+^ and Iba1^+^ cells [%], *p* = 0.059; Figure 9H).

Complementary to the histological examinations, RT-qPCR analyses for the different subunits of C1q-complex, namely *C1QA*, *C1QB*, and *C1QC*, were performed at two and four days of cultivation. At two days, mRNA levels of *C1QA* were significantly upregulated through oxidative stress induction (1.69-fold expression, *p* = 0.028), whereas the expression of *C1QA* remained unchanged in CoCl_2_ group (1.29-fold expression, *p* = 0.396). In contrast, after four days, the regulation of *C1QA* mRNA was not altered through H_2_O_2_ or CoCl_2_ (H_2_O_2_: 0.88-fold expression, *p* = 0.695; CoCl_2_: 0.55-fold expression, *p* = 0.051; Figure 9I). *C1QB* mRNA levels were not changed after oxidative stress or hypoxia at two days (H_2_O_2_: 1.42-fold expression, *p* = 0.271; CoCl_2_: 1.10-fold expression, *p* = 0.862). The same was observed at four days (H_2_O_2_: 1.15-fold expression, *p* = 0.607; CoCl_2_: 0.87-fold expression, *p* = 0.486; Figure 9J). Oxidative stress, due to H_2_O_2_, had no effect *C1QC* expression at two days (1.10-fold expression, *p* = 0.723). In contrast, hypoxic processes, induced by CoCl_2_, led to a significant downregulation of *C1QC* at two days (CoCl_2_: 0.55-fold expression, *p* = 0.046). At four days, *C1QC* expression was similar in all three groups (H_2_O_2_: 0.83-fold expression, *p* = 0.571; CoCl_2_: 0.56-fold expression, *p* = 0.090; Figure 9K).

Microglia are producers of several inflammatory cytokines. To investigate the inflammation in retinas after oxidative stress and hypoxia, RT-qPCR analyses of *tumor necrosis factor* α (*TNF*) as well as *interleukin 6* (*IL6*) and *8* (*IL8*) were performed. Even though at two days, the mRNA expression of *TNF* was twice as high through H_2_O_2_ this effect was not significant (1.94-fold expression, *p* = 0.068). In contrast, hypoxic processes due to CoCl_2_ had no effect on the expression level of *TNF* (1.23-fold expression, *p* = 0.400). However, at four days, *TNF* mRNA expression was 3.85-fold upregulated in H_2_O_2_-stressed retinas (*p* = 0.002). CoCl_2_ still had no effect on the regulation of *TNF* (1.11-fold expression, *p* = 0.836; Figure 9L). *IL6* mRNA expression was significantly upregulated due to H_2_O_2_ at two days (3.46-fold expression, *p* = 0.021), whereas the expression of *IL6* remained unaltered in CoCl_2_ retinas (0.79-fold expression, *p* = 0.085). At four days, a significantly upregulated mRNA expression of *IL6* was also noted in H_2_O_2_-stressed retinas (2.57-fold expression, *p* = 0.023). In CoCl_2_-stressed retinas, the *IL6* expression was twice as high, but this effect was not significant (1.82-fold expression, *p* = 0.224; Figure 9M). At two days, analysis of *IL8* mRNA expression revealed a 3.74-fold upregulation in the H_2_O_2_ group (*p* = 0.060). In CoCl_2_ retinas, the *IL8* level was not altered (0.72-fold expression, *p* = 0.630). However, at four days, both stressors, H_2_O_2_ as well as CoCl_2_, led to a significant upregulation of *IL8* mRNA levels (H_2_O_2_: 4.93-fold expression, *p* = 0.036; CoCl_2_: 3.54-fold expression; *p* = 0.033; Figure 9N).

## 4. Discussion

For numerous retinal diseases, such as glaucoma, AMD, or diabetic retinopathy, there are still no appropriate universal therapeutics available for healing patients. Current treatments mainly focus on slowing down disease progression, e.g., in glaucoma by decreasing the intraocular pressure. In addition, for NGV a secondary form glaucoma, anti-VEGF agents are also commonly used [2].

It is known that oxidative stress or hypoxia play a role in these retinal diseases [46,47]. For in vitro evaluations, pathomechanisms, such as oxidative stress and hypoxia, can be induced chemically. H_2_O_2_, as a strong oxidizer of the ROS family, is formed as a byproduct during the respiratory chain or cell metabolisms, such as protein folding. Among other functions, H_2_O_2_ is involved in the immune response. Too high levels of H_2_O_2_, that cannot be reduced by antioxidants, lead to cell cycle arrest, inflammation, and apoptosis [48]. In contrast, hypoxia precedes the transcription of several genes, which are of high importance for tissue protection and adaption, as for angiogenesis or metabolism. Moreover, the dampening of inflammation processes occurs under hypoxia. The adaption of the tissue to oxygen deprivation, such as anaerobic ATP synthesis, is performed by genes of metabolism, such as phosphoglycerate kinase and lactate dehydrogenase A. The improvement of oxygen supply is carried out via the transcription of erythropoietin and VEGF. Furthermore, hypoxia triggers the dampening of inflammation via the inhibition of nucleoside transporters and the increase of extracellular adenosine levels [49].

Increasing numbers of ex vivo organ culture studies indicate that these models are crucial for retinal research and increasingly more come to the fore [30,50,51]. In contrast to cell cultures, organ culture models have the advantage of an intact organ, such as the retina, mimicking in vivo situation more closely. Furthermore, no euthanasia is necessary for just these experiments, since pig eyes can be obtained from local slaughterhouses where they are a waste product of the food industry. To this end, the organ culture model of porcine retinas is an excellent alternative to cell culture as well as to animal models. Cellular processes and cell–cell interactions, such as the role of the complement system, can be effectively investigated in damaged porcine retinas.

Several studies have shown that H_2_O_2_ as well as CoCl_2_ induce strong neurodegenerative effects in the retina, leading to a loss of RGCs [6,7,29,36].

In the present study, we used both substances to induce neurodegenerative pathomechanisms in cultured porcine retinas to investigate the role of the complement cascade in this degeneration model. The complement system, as a part of the innate immune system, is known to be involved in complex neurodegenerative diseases such as glaucoma [13,41,52]. In the present study, we showed that hypoxic stress triggered by CoCl_2_ induced a loss of retinal neurons by apoptosis. Furthermore, CoCl_2_ led to the activation of the complement system via the classical pathway, were C1q-expressing microglia were significantly upregulated despite the toxicity of CoCl_2_ for microglia. However, the complement system did not seem to play such a crucial role in oxidative stressed retinas, here we rather noted strong inflammation due to an increased mRNA expression of *TNF*, *IL6*, and *IL8*. To this end, the present study undermines the role of the complement system in an alternative retinal degeneration model of cultivated porcine retinas, stressed with H_2_O_2_ or CoCl_2_.

Retinal thinning as well as thinning of the ganglion cell complex (GCL, IPL, and INL) was seen after the induction of hypoxic processes through CoCl_2_. This thinning of CoCl_2_-damaged porcine retinas is in accordance with recent studies by our group [6,29]. Also, rats and mice that underwent ischemia, show a significant loss of retinal thickness already after three days and a significant thinning of the GCL as early as after six hours [45,53]. Mesentier-Louro et al. recently observed that a systemic induction of hypoxia through 10% O_2_ leads to a thinning of the retina in 6- to 8-weeks-old mice. Hypoxic stress reduced the metabolic support of RGC axons by damaging oligodendrocytes in the optic nerve [54]. In our model, we detected a high sensitivity of RGCs to oxidative stress as well as hypoxia since significant RGC loss through apoptosis was noted. The loss of RGCs might lead, among other factors, to a reduced retinal thickness. Usually, a balance of anti-apoptotic and pro-apoptotic proteins are crucial for a cell survival [55]. We noted that H_2_O_2_ as well as CoCl_2_ induce a disbalance of Bax, a pro-apoptotic protein, and Bcl-2, a pro-survival protein [56], in this organ culture model. Taken together, hypoxic as well as oxidative stressed porcine retinas demonstrated a disrupted balance, where cell survival-especially of RGCs-cannot be assured.

In general, the complement system can be activated through different routes. These three pathways result in the activation of C3 and further downstream proteins, such as C5 and MAC [43]. It is postulated that the complement system contributes to retinal diseases, such as glaucoma [5,41,57]. Hence, we investigated the role of the complement system in cultivated porcine retinas that were exposed to oxidative stress or hypoxia, two pathomechanisms involved in many retinal diseases. In oxidative stressed porcine retinas, no complement system activation of the terminal pathway occurred. Nevertheless, an early loss of RGCs due to apoptotic processes was observed. A dysregulated complement system was seen by an increased mRNA expression of *CFH*, even though no complement activation was found. However, in oxidative stressed retinas, strong inflammation processes were observed in form of proliferating microglia and increased mRNA expression of genes encoding for inflammatory cytokines such as *TNF*, *IL6*, and *IL8*.

On the other hand, we noted an involvement of the complement system in hypoxic damaged porcine retinas. Hypoxia, induced by CoCl_2_, led to a strong activation of the terminal pathway, especially increased numbers of C3^+^ cells after two days and more MAC^+^ cells after four days. C3, as a member of the complement cascade, is strongly involved in glaucoma disease [13,58]. In line with these findings, treatment with an adenovirus inhibiting C3 in DBA/2J mice, which develop a secondary form of glaucoma, led to a neuroprotection of RGCs and their axons [59]. Furthermore, the intravitreal inhibition of C5, which is a downstream-protein of the terminal complement cascade and activated by C3, prevented RGC loss and preserved retinal function in a rat autoimmune glaucoma model [60].

Moreover, a connection between the complement system and apoptosis is discussed. Jha et al. reported a complement mediated loss of RGCs via apoptosis [61]. In our hypoxia induced degeneration model of porcine retinas, we observed similar effects. Hypoxia induced a very early complement activation, at day two, whereas RGCs loss via apoptosis occurred after four days. Together, the complement system might amplify retinal damage leading to an increased RGC loss. In the present study, hypoxia due to CoCl_2_ triggered complement system activation via the classical pathway, where significantly more microglia were C1q^+^ even though CoCl_2_ is toxic to microglia and induces their loss.

Glia, such as microglia, are a main source of C1q in the brain but also in the developing retina [62,63]. Our colocalization staining further confirmed that retinal microglia express C1q. It is discussed that ischemic ganglion cells bind C1q and therefore induce the activation of the complement cascade [13,64]. Furthermore, it is evidenced that C1q binds to dying neurons, RGCs, and synapses and therefore leads to the activation of the classical pathway of the complement cascade [52,65]. A recent study shows that in the developing retina of mice C1q is mainly expressed by microglia and very important for the regulation of horizontal cell neurite confinement. Furthermore, they noted that when C1q is lacking, microglia activation is decreased [63].

However, in our study, we see that C1q is mainly present in inner retinal layers and not as excessively in the GCL. Elevated C1q levels were previously reported by several studies in regard to glaucomatous retinas [13,57]. A study by Stasi et al. describes increased levels of C1q mRNA in retinas of old DBA/2J glaucomatous mice. Moreover, in laser-induced glaucomatous monkey eyes, more C1q in the retina was noted [57].

Our study reveals a connection between C1q and microglia in the hypoxia mediated activation of the complement system. The exact role of C1q expressing microglia is not sufficiently understood. It is discussed that C1q expressing microglia are an instigator for inflammatory processes leading to increased tissue damage. In line with this, Jiao et al. observed that C1q deficient mice have not only fewer microglia but also show less retinal damage [66]. Hence, C1q expressing microglia might contribute to additional retinal damage also in our porcine model.

## 5. Conclusions

This is the first study where the relationship between oxidative stress or hypoxia and the complement system was investigated in a degeneration model of porcine retinas. Here, oxidative stress did not activate the complement system, but induced strong inflammation processes caused by proliferating microglia, leading to RGC loss, and undergoing apoptosis. In contrast, hypoxia, induced by CoCl_2_, activated the complement cascade via the classical pathway and time-dependently triggered the complement system, leading to a C3 and MAC upregulation. Hence, we conclude that there might be a connection between the complement cascade and ongoing apoptotic processes leading to RGC loss. Furthermore, an involvement of C1q expressing microglia likely contributes to retinal damage. Together, this study indicates that the porcine retinal organ culture model is an excellent alternative model to study the role of pathomechanisms and the complement system in retinal diseases (Figure 10).

## Figures and Tables

**Figure 1 cells-10-03575-f001:**
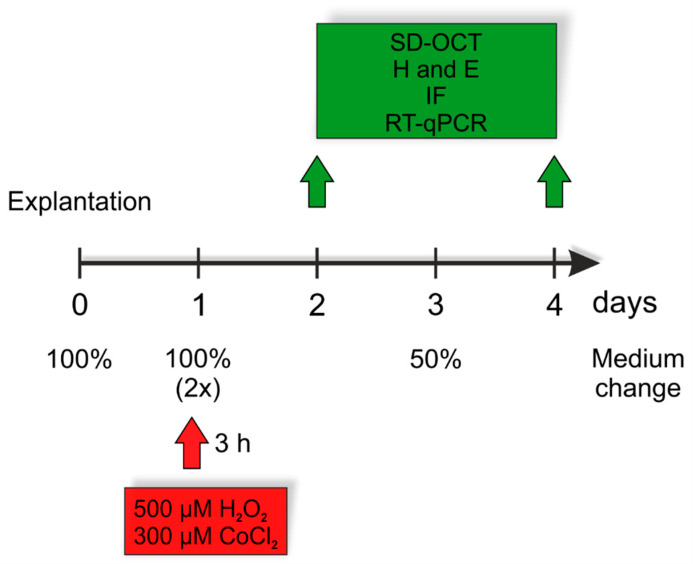
Experimental setup. Porcine retinas were cultivated for two and four days. Degeneration was induced by H_2_O_2_ (500 µM) or CoCl_2_ (300 µM) at day one for three hours. Control retinas were cultivated without a stressor. Each group received a complete medium change on days zero and one. Half of the medium was renewed on day three. On day two and four, spectral domain-optical coherence tomography (SD-OCT), hematoxylin and eosin staining (H & E), and immunofluorescence (IF) as well as quantitative real-time PCR (RT-qPCR) analyses were performed.

**Figure 2 cells-10-03575-f002:**
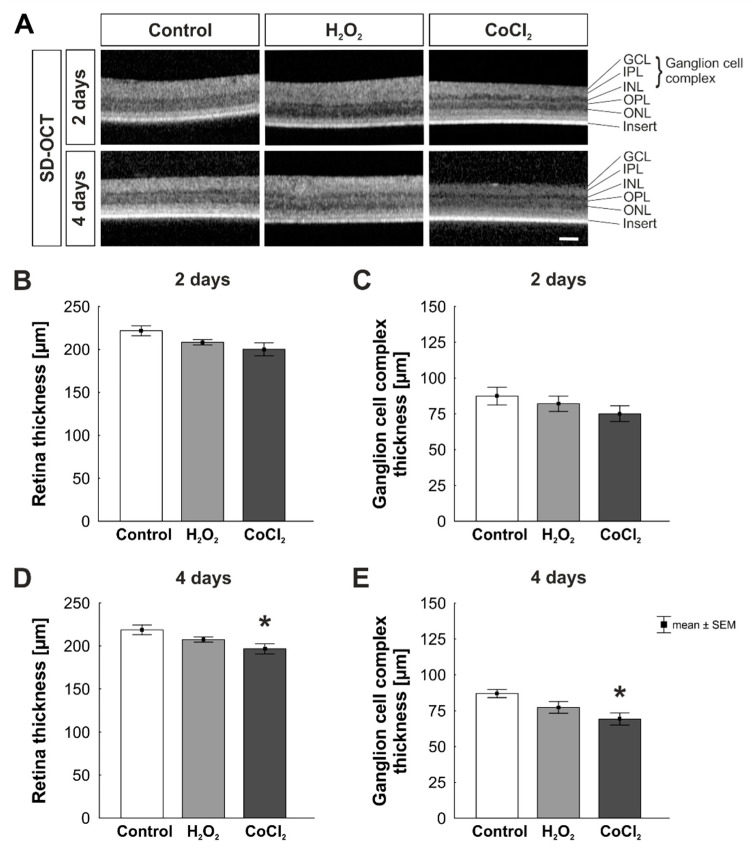
Loss of retinal thickness caused by hypoxia. (**A**) Exemplary SD-OCT-images of retinas cultivated for two and four days. (**B**) Total retinal thickness was not affected by oxidative stress nor hypoxia after two days. (**C**) The thickness of the ganglion cell complex (GCL + IPL + INL) was not altered in any of the groups after two days. (**D**) A significantly thinner retina thickness was noted in hypoxia-stressed retinas at four days (*p* = 0.024). (**E**) Also, the ganglion cell complex of hypoxia stressed retinas was significantly thinner than control retinas (*p* = 0.015). GCL = ganglion cell layer; IPL = inner plexiform layer; INL = inner nuclear layer; OPL = outer plexiform layer; ONL = outer nuclear layer. N = 5–6. Scale bar: 100 µm. Values are shown as mean ± SEM. Columns marked with * show significant differences in comparison to the control group. * *p* < 0.05.

**Figure 3 cells-10-03575-f003:**
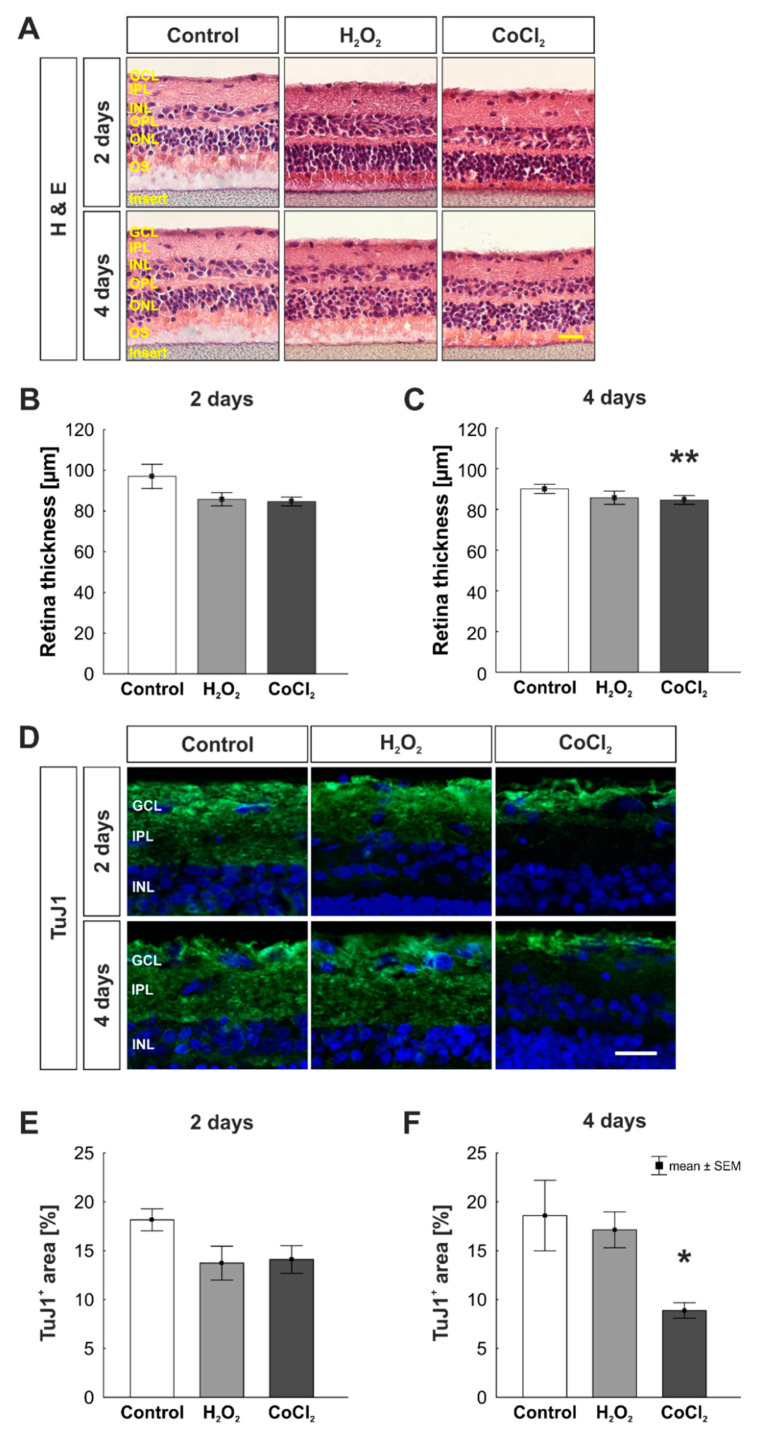
Reduced retinal thickness through hypoxic stress. (**A**) Exemplary images of H & E-stained retinas at two and four days. (**B**) No alteration regarding the retinal thickness was noted within the groups at two days. (**C**) At four days, significantly thinner retinas were observed in the hypoxia-stressed group (*p* = 0.006). (**D**) Representative images of TuJ1 (green) stained cross-sections. DAPI was used to visualize cell nuclei (blue). (**E**) Area measurements of immunopositive area revealed no alteration within the groups at two days. (**F**) At four days, a significant reduction of TuJ1^+^ area in CoCl_2_-stressed retinas was noted (*p* = 0.029). GCL = ganglion cell layer; IPL = inner plexiform layer; INL = inner nuclear layer; OPL = outer plexiform layer; ONL = outer nuclear layer; OS = outer segments. (**B**,**C**): n = 8; (**E**,**F**): n = 6. Scale bars: 20 µm. Values are shown as mean ± SEM. Columns marked with * show significant differences in comparison to the control group. * *p* < 0.05; ** *p* < 0.01.

**Figure 4 cells-10-03575-f004:**
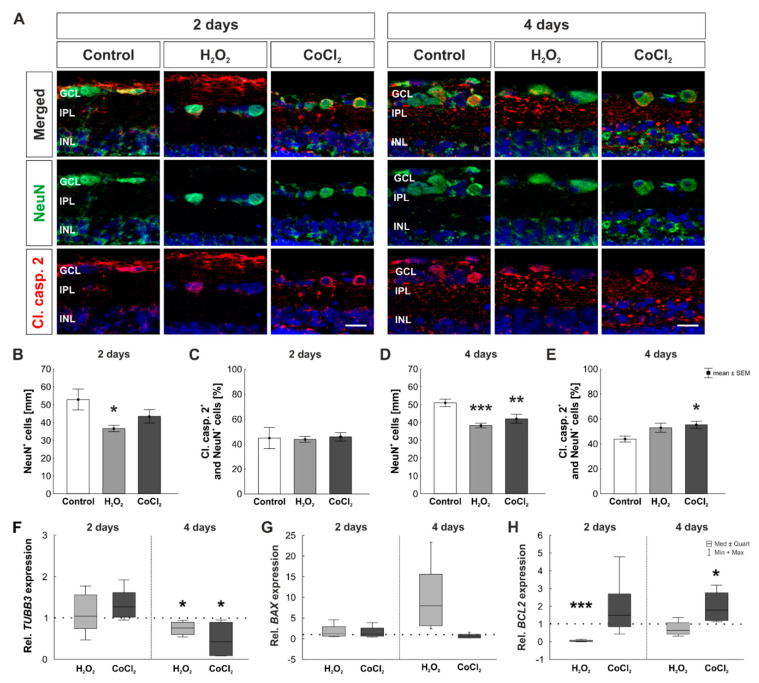
Loss of retinal neurons due to oxidative stress and hypoxia. (**A**) Retinal neurons were stained with a NeuN antibody (green) and co-stained with the apoptotic marker cleaved caspase 2 (red). Cell nuclei were labelled with DAPI (blue). (**B**) A significant loss of NeuN^+^ cells was noted in the H_2_O_2_ group (*p* = 0.04), but not in the CoCl_2_ group after two days. (**C**) At day two, the number of apoptotic NeuN^+^ cells was comparable in all three groups. (**D**) At four days, both stressors, H_2_O_2_ and CoCl_2_, led to a significant loss of NeuN^+^ cells (*p* < 0.001 and *p* = 0.011). (**E**) NeuN^+^ cells loss was accompanied by an increased number of apoptotic cells after CoCl_2_-stress (*p* = 0.035). (**F**) Significant downregulation of *TUBB3* mRNA was noted in both stressor groups after four days (both: *p* = 0.02). (**G**) mRNA expression levels of *BAX* was not altered between groups at both investigated points in time. (**H**) H_2_O_2_ led to a significant downregulation of *BCL2* mRNA expression at two days (*p* = 0.002), *BCL2* levels were significantly upregulated in CoCl_2_ samples at four days (*p* = 0.03). GCL = ganglion cell layer; IPL = inner plexiform layer; INL = inner nuclear layer. (**B**,**C**): n = 5; D, (**E**): n = 8; (**F**–**H**): n = 4–5. Scale bars: 20 µm. Values are shown as mean ± SEM for immunofluorescence and median ± quartile + min/max for RT-qPCR. The dotted lines in (**F**–**H**) represent the relative expression of the control group. Columns marked with * show significant differences in comparison to the control group. * *p* < 0.05; ** *p* < 0.01; *** *p* < 0.001.

**Figure 5 cells-10-03575-f005:**
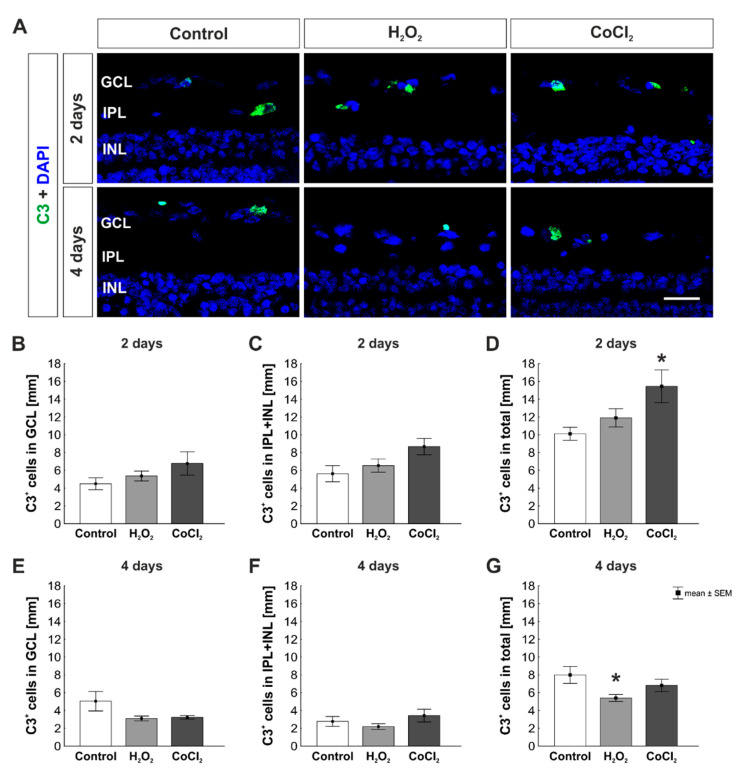
Early activation of the complement cascade in hypoxic retinas. (**A**) Cells were labeled with a C3 antibody (green) and cell nuclei with DAPI (blue). (**B**) The number of C3^+^ cells located in the GCL was comparable in all groups at two days. (**C**) The same was seen for C3^+^ cells located in the IPL and INL. (**D**) The total amount of C3^+^ cells in hypoxic retinas was significantly increased at two days (*p* = 0.021). (**E**) At four days, the number of C3^+^ cells in the GCL was comparable in all groups. (**F**) Moreover, the C3^+^ cell count in the IPL and INL was not altered by H_2_O_2_ or CoCl_2_. (**G**) Interestingly, in the total retina, there were significantly fewer C3^+^ cells in oxidative stressed retinas (*p* = 0.046). GCL = ganglion cell layer; IPL = inner plexiform layer; INL = inner nuclear layer. N = 8. Scale bar: 20 µm. Values are shown as mean ± SEM. Columns marked with * show significant differences in comparison to the control group. * *p* < 0.05.

**Figure 6 cells-10-03575-f006:**
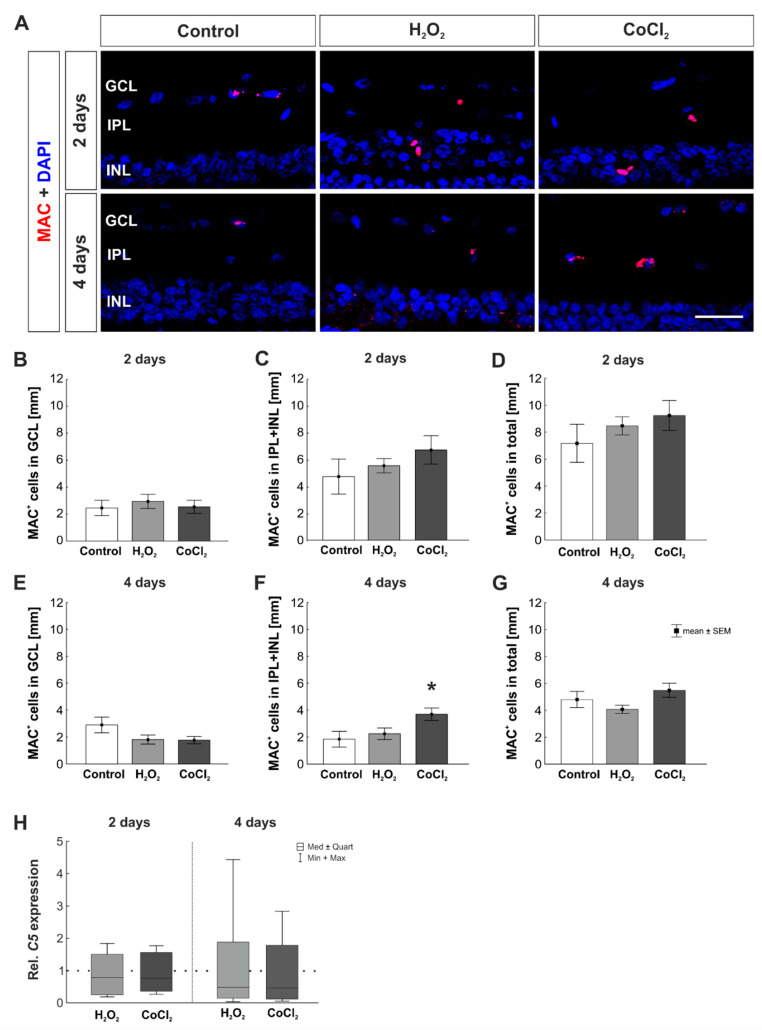
Late activation of MAC through CoCl_2_ in the inner retinal layers. (**A**) Antibody against MAC was used to visualize MAC^+^ cells in the retina (red). Cell nuclei were labeled with DAPI and are shown in blue. (**B**) Neither H_2_O_2_ nor CoCl_2_ had any effect on the number of MAC^+^ cells located in the GCL at two days. (**C**) At two days, the number of MAC^+^ cells was not altered by H_2_O_2_ or CoCl_2_ in inner retinal layers. (**D**) In the total retina, similar MAC^+^ cell counts were detected in all retinas. (**E**) At four days, the number of MAC^+^ cells in the GCL was alike in all three groups. (**F**) Interestingly, at four days, significantly more MAC^+^ cells were counted in CoCl_2_ retinas in IPL and INL (*p* = 0.039). (**G**) This effect was not seen, when counting MAC^+^ cells in the total retina. GCL = ganglion cell layer; IPL = inner plexiform layer; INL = inner nuclear layer. (**B**–**G**): n = 8; (**H**) n = 4. Scale bar: 20 µm. Values are shown as mean ± SEM for immunofluorescence and median ± quartile+min/max for RT-qPCR. The dotted line in H represents the relative expression of the control group. Columns marked with * show significant differences in comparison to the control group. * *p* < 0.05.

**Figure 7 cells-10-03575-f007:**
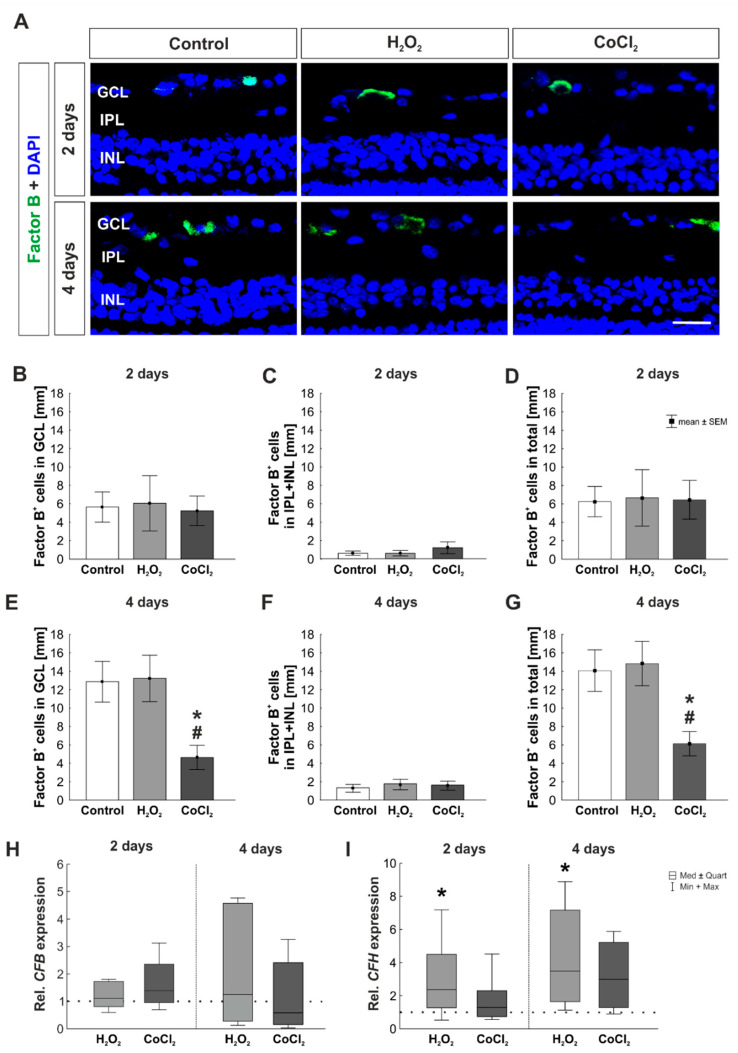
No activation of the complement system via the alternative pathway. (**A**) Retinal cross-sections were stained with an anti-factor B antibody (green), to investigate the alternative pathway. Cell nuclei (DAPI) are shown in blue. (**B**) At two days, neither H_2_O_2_ nor CoCl_2_ had any effect on the number of factor B^+^ cells in the GCL. (**C**) Additionally, in the IPL and INL, no changes were noted in factor B^+^ cell counts through H_2_O_2_ or CoCl_2_. (**D**) In accordance, in the total retina the factor B^+^ cells were unaltered. (**E**) Interestingly, CoCl_2_ led to a significant loss of factor B^+^ cells in the GCL at four days (*p* = 0.02). (**F**) This effect was not seen in the IPL and INL. (**G**) Regarding cell counts of factor B in the total retina, a significant loss was seen in CoCl_2_-stressed retinas at four days (*p* = 0.03). (**H**) mRNA expression levels of *CFB* were not altered at any point in time. (**I**) However, *CFH* mRNA levels were significantly upregulated in H_2_O_2_-stressed retinas at both investigated points in time (*p* = 0.04 and *p* = 0.01). GCL = ganglion cell layer; IPL = inner plexiform layer; INL = inner nuclear layer. (**B**–**G**): n = 8; (**H**,**I**): n = 4–5. Scale bar: 20 µm. Values are shown as mean ± SEM for immunofluorescence median ± quartile +min/max for RT-qPCR. The dotted lines in H and I represent the relative expression of the control group. Columns marked with * show significant differences in comparison to the control group. Columns marked with ^#^ show significant difference between H_2_O_2_ and CoCl_2_. *,^#^
*p* < 0.05.

**Figure 8 cells-10-03575-f008:**
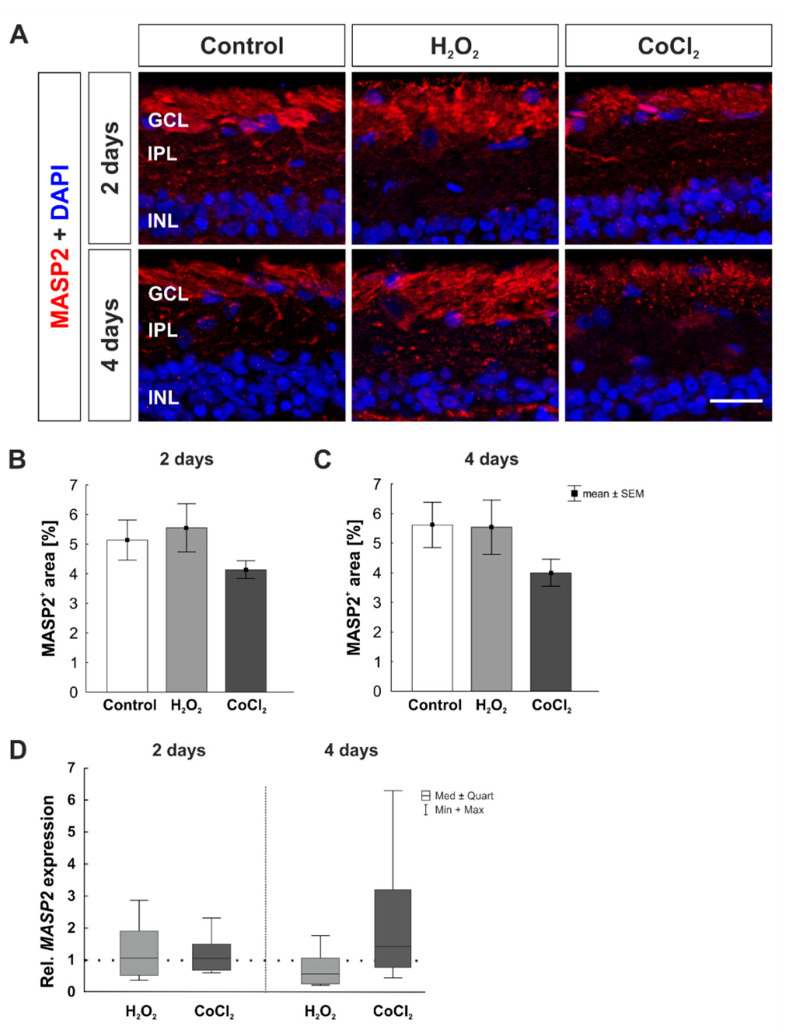
No activation of the complement system via the lectin pathway. (**A**) For investigation of the lectin pathway, a MASP2 antibody were used (red). Cell nuclei were stained with DAPI (blue). (**B**) MASP2^+^ staining area was not altered in any of the groups at two days. (**C**) Additionally, at four days, no changes were found in MASP2^+^ signal area. (**D**) In accordance, mRNA expression levels of *MASP2* remained unchanged after H_2_O_2_ or CoCl_2_ application. GCL = ganglion cell layer; IPL = inner plexiform layer; INL = inner nuclear layer. B, C: n = 8; D: n = 4–5. Scale bar: 20 µm. Values are shown as mean ± SEM for immunofluorescence median ± quartile +min/max for RT-qPCR. The dotted line in D represents the relative expression of the control group.

**Figure 9 cells-10-03575-f009:**
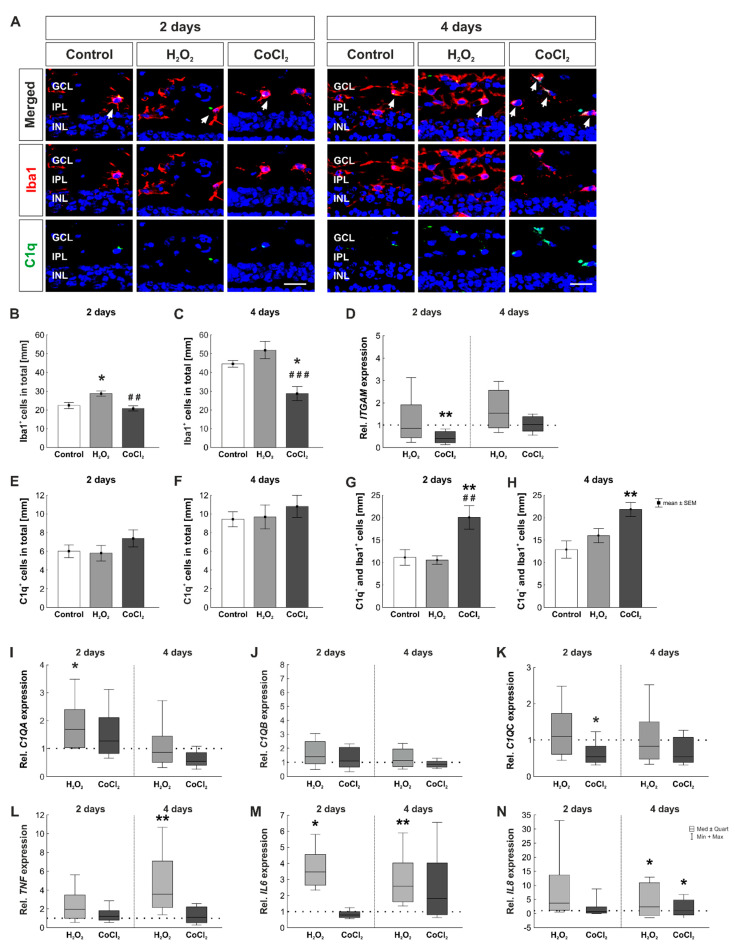
Less microglia but more C1q^+^ microglia in hypoxic retinas and increased inflammation in oxidative stress retinas. (**A**) In order to investigate microglia and their C1q immunoreactivity, a double staining with Iba1 (microglia, red) and C1q (green) was performed at two and four days. Arrows show co-localized cells. DAPI was used to mark cell nuclei (blue). (**B**) Oxidative stress induced a significantly increased number of microglia at two days (*p* = 0.016). (**C**) At four days, CoCl_2_-stressed retinas showed a significant loss of microglia (*p* = 0.014). (**D**) mRNA expression of *ITGAM*, a gene which encodes for Cd11b, was significantly downregulated in CoCl_2_ retinas at two days (*p* = 0.003). (**E**) At two days, the total number of C1q^+^ cells in the retina was not altered in any group. (**F**) Also, at four days, the C1q^+^ cell counts were comparable in all groups. (**G**) At two days, CoCl_2_-stressed retinas had significantly more C1q^+^ microglia than H_2_O_2_-retinas (*p* = 0.009) as well as control retinas (*p* = 0.005). (**H**) After four days of cultivation, CoCl_2_-stressed retinas displayed significantly more C1q^+^ microglia than control retinas (*p* = 0.003). (**I**) *C1QA* mRNA levels were significantly upregulated in H_2_O_2_-stressed retinas at two days (*p* = 0.03). (**J**) RT-qPCR analyses revealed an unaltered mRNA expression level of *C1QB* in H_2_O_2_- and CoCl_2_-stressed retinas in both investigated points in time. (**K**) Regarding *C1QC* mRNA levels, a significant downregulation was noted in hypoxic retinas at two days (*p* = 0.046). (**L**) *TNF* mRNA levels were significantly upregulated after four days, when retinas were exposed to oxidative stress (*p* = 0.002). (**M**) Retinas, which were cultivated with H_2_O_2_, had a significantly upregulated *IL6* mRNA expression at two and four days (both: *p* = 0.02). (**N**) Oxidative stress had no effect on the mRNA expression of *IL8* at two days and a significant upregulation at four days (*p* = 0.04). Hypoxic processes also increased the mRNA expression of *IL8* significantly at four days (*p* = 0.03). GCL = ganglion cell layer; IPL = inner plexiform layer; INL = inner nuclear layer. (**B**,**C**,**E**–**H**): n = 8; (**D**,**I**–**N**): n = 4–5. Scale bar: 20 µm. Values are shown as mean ± SEM for immunofluorescence and median ± quartile +min/max for RT-qPCR. The dotted lines in (**D**,**I**–**N**) represent the relative expression of the control group. Columns marked with * show significant differences in comparison to the control group. Columns marked with ^#^ show significances between H_2_O_2_ and CoCl_2_. * *p* < 0.05; **, ^##^
*p* < 0.01; ^###^
*p* < 0.001.

**Figure 10 cells-10-03575-f010:**
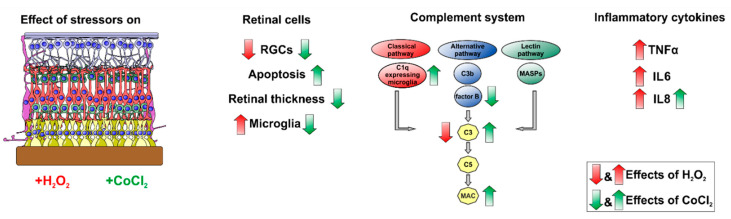
Graphical summary of the study. Porcine retinas were cultivated for two and four days. The role of the complement system was evaluated after the induction of oxidative stress or hypoxia. Both stressors, H_2_O_2_ (shown with red arrows) and CoCl_2_ (shown with green arrows) had strong degenerating effects on retinal ganglion cells (RGCs). Hypoxic retinas, stressed with CoCl_2_, had more apoptotic RGCs and had a significantly decreased thickness of the whole retina as well as of the ganglion cell complex. Different effects of H_2_O_2_ and CoCl_2_ on microglia were noted: H_2_O_2_-stressed retinas led to more microglia whereas CoCl_2_ was toxic to microglia. The complement system played no prominent role in oxidative stressed retinas, but there was a strong activation of the complement system occurring via the classical pathway in hypoxic retinas. Furthermore, we noted a strong increase of inflammatory cytokines mostly in H_2_O_2_-stressed retinas.

**Table 1 cells-10-03575-t001:** Primary and secondary antibodies used for immunofluorescence.

Primary Antibodies				Secondary Antibodies		
Antibody	Source	Company	Dilution	Antibody	Company	Dilution
Anti-C3	Rabbit	Cedarlane	1:500	Goat anti-rabbit Alexa Fluor 488	Invitrogen	1:500
Anti-cleaved caspase 2	Rabbit	Abcam	1:300	Donkey anti-rabbit Alexa Fluor 555	Invitrogen	1:500
Anti C5b-9 (MAC)	Mouse	Biozol	1:100	Donkey anti-mouse Alexa Fluor 555	Abcam	1:500
Anti-Iba1	Chicken	Synaptic Systems	1:500	Donkey anti-chicken Cy3	Millipore	1:500
Anti-C1q	Goat	Quidel	1:500	Donkey anti-goat Alexa Fluor 488	Dianova	1:500
Anti-MASP2	Rabbit	Biozol	1:200	Donkey anti-rabbit Alexa Fluor 555	Invitrogen	1:700
Anti-factor B	Goat	Tecomedial	1:1000	Donkey anti-goat Alexa Fluor 488	Dianova	1:500
Anti-NeuN	Chicken	Millipore	1:500	Donkey anti-chicken Alexa Fluor 488	Jackson Immuno Research	1:500
Anti-TuJ1	Mouse	Covance	1:300	Donkey anti-mouse Alexa Flour 488	Invitrogen	1:500

**Table 2 cells-10-03575-t002:** Primer pairs used for RT-qPCR.

Gene	Primer Forward (5′–3′)	Primer Reverse (5′–3′)
*C1QA*	*CGACAGAATCCTCCGACGAG*	*GCTGGACCTGGTCTCTCCTA*
*C1QB*	*TCAAGGGAGAGAAAGGGTTGC*	*AAGTAGTAAAGGCCGGGCAC*
*C1QC*	*TCCTGGCCCCTTCTGGTACT*	*GTAGTAGAGGCCGGGGACTT*
*TUBB3*	*CAGATGTTCGATGCCAAGAA*	*GGGATCCACTCCACGAAGTA*
*BAX*	*GGACCATCGGTATTGGTGTC*	*AGATGAGGGAGAGAGGCACA*
*BCL2*	*GCTCGTGCGGGATTGACTACTACA*	*CCAGCGGGTTCTTGCCACAGC*
*C5*	*ACTTGGTGACCTTCGACGTG*	*ACCCCTTGGGTCCAGAGTAA*
*CFB*	*CTCAACGCAAAGACCGCAAA*	*AAATGGGCCTGATGGTCTGG*
*CFH*	*TATCCTCCGGGAACAGTCGT*	*ACTTTGCCTTGCTGACAGGT*
*MASP2*	*GGCTTCCCCGAAAAGTATGC*	*GGGTGAAGTAGAGACGCAGG*
*TNF*	*GCCCTTCCACCAACGTTTTC*	*CAAGGGCTCTTGATGGCAGA*
*IL6*	*GCAGTCACAGAACGAGTGGA*	*CTCAGGCTGAACTGCAGGAA*
*IL8*	*TTCCAAACTGGCTGTTGCCT*	*ACAGTGGGGTCCACTCTCAA*
*ACTB*	*CTCTTCCAGCCTTCCTTC*	*GGGCAGTGATCTCTTTCT*
*H3*	*ACTGGCTACAAAAGCCGCTC*	*ACTTGCCTCCTGCAAAGCAC*

## Data Availability

Data sharing not applicable.

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
