# Peer review of "Hypoxic Processes Induce Complement Activation via Classical Pathway in Porcine Neuroretinas"

_cells, 2021, doi:10.3390/cells10123575_

Round 1

Reviewer 1 Report

The study entitled “Hypoxic processes induce complement activation via classical 2 pathway in porcine neuroretinas “performed by Mueller-Buehl et al., investigated the molecular mechanism of associated with neuronal damage in retinal diseases. Authors utilized an organ culture model, porcine retinal explants treated with H2O2 or CoCl2 as the experimental model, and investigated the relation between oxidative stress /hypoxia and the complement system was investigated. Lack of suitable experimental models is a challenge in the field of vision; this study provides a great platform for investigating the underlying molecular mechanisms in various retinal diseases. However, the study needs further clarification at several stages before the results can be concluded.

  1. Under abstract and introduction, authors project this study as a model investigating the molecular mechanisms associated with glaucoma. However, in the conclusion section, they state this porcine retinal organ culture model is an excellent alternative model to study the role of pathomachanisms and the complement system in retinal diseases. Considering the fact that neuronal damage, excitotoxicity, inflammation, complement activation etc. are observed in multiple retinal diseases, authors should consider revising the abstract and introduction with a more general approach.
  2. Figure 4. Representation of statistical significance is confusing. It would be better if the “*” are placed above the histograms representing CoCl2 treatment showing reduction, and describe clearly under the figure legends. It would be great if the authors can follow similar pattern in other figures as well.
  3. Figure 3. Panel 3A images are of good quality, but appear smaller to visualize the details. Readers will benefit if the images can be can be enlarged.
  4. Fig 3, cleaved caspase 3 is showing staining in IPL and INL. Did the authors try to see what this could be? No cell bodies are expected to be present in IPL. Are there evidence of cell death in other neuronal layers?
  5. It would be ideal to confirm RGC damage using an additional marker such as Tuj1.
  6. Fig 4 and 5, the images presented do not represent the quantification data provided. Only one or two cells are present per FOV and no difference across the groups. Please present representative images covering more area of the retina to demonstrate the results.
  7. Fig 6, panels A and J need to be enlarged to see the details of the immunostaining results.
  8. Quantification method adapted for counting cells or measuring areas in this study need to be described with all the details under the methods section. As of now, the Y-axes do not provide sufficient information. Number of sections quantified per animal need to be included. What is meant my “mm” as shown in many graphs? How this quantification was performed needs to be described.
  9. Fig 6, and other figures, expression of RNA markers in control groups are not shown. If the comparisons are made with respect to control, authors need to state in clearly under figure legends.
  10. Fig 7B and C, it is not clear whether the changes in microglia in CoCl2 treated group is compared to H2O2 treated group or control group. In fact, the comparison should be done to control group. Results say, “induced a significant loss of microglia.” However, it appears like the number of microglia stay the same in control and CoCl2 treated groups on 2 and 4 days. This need to be addressed.

Reviewer 2 Report

The paper entitled “Hypoxic processes induce complement activation via classical pathway in porcine neuroretinas” by Mueller-Buehl deals with the study of the retinal inflammation and in particular the role of the complement system and microglia in hypoxic and oxidative stressed cultured porcine retinas.

Although the paper addresses an issue of interest in the field, the authors may wish to consider the following prior to publication.

Page 1 line 37: the authors wrote “Increased intraocular pressure  (IOP) is the main risk factor, but there are many studies which examined the role of other possible pathogenic factors like excitotoxicity, circulatory disorders, oxidative stress, hypoxia, a dysregulated immune system and more”, the authors should add that  “Almost all the forms of glaucoma are associated with elevated intraocular pressure (IOP), including the neovascular glaucoma (NGV) that is characterized by inflammation and high levels of VEGF in the eye. For NGV the therapeutic approach could be different, on this regards it is worth of note that anti-VEGF agents, used in clinical practice, such as ranibizumab, bevacizumab and aflibercept acting also against inflammation throughout PLA2/COX-2/VEGF-A pathway (please add this relevant paper on reference section Biochem Pharmacol. 2015 Aug 1;96(3):278-87. doi: 10.1016/j.bcp.2015.05.017. Epub 2015 Jun 6. PMID: 26056075). Incidentally, the authors used a chemical ischemia model (cobalt chloride model) in the study, the authors should underline that CoCl2 can modulate HIF-1α, VEGF, PlGF (Front Pharmacol. 2020 Jul 17;11:1063.  doi: 10.3389/fphar.2020.01063). Please modify this sentence widening this point and helping the reader that is not familiar with the field.

Discussion Section: the authors claimed that “For numerous retinal diseases, such as glaucoma, there are still no appropriate therapeutics available to heal patients. Current treatments can only slow down disease progression”, the authors should expand this point and add that the current pharmacological therapies act against the increase of IOP and for NGV are also available the anti-VEGF. The authors also claimed that “hypoxia precedes the transcription of several 487 genes, inducing inflammation, angiogenesis, or metabolism” this is true and the authors should emphasize this point for the benefit of the reader.

Round 2

Reviewer 1 Report

The authors have addressed all the comments adequately and the manuscript is in acceptable form.

Author Response

We kindly thank the reviewer for the comment. We carefully revised the manuscript for typos and spell check.

Reviewer 2 Report

the new version has been improved, the paper can be published now.

Check some typo throughout the text.

Author Response

We thank the reviewer for this comment. We revised our manuscript very carefully for typos.